# Oralbiotica/Oralbiotics: The Impact of Oral Microbiota on Dental Health and Demineralization: A Systematic Review of the Literature

**DOI:** 10.3390/children9071014

**Published:** 2022-07-08

**Authors:** Alessio Danilo Inchingolo, Giuseppina Malcangi, Alexandra Semjonova, Angelo Michele Inchingolo, Assunta Patano, Giovanni Coloccia, Sabino Ceci, Grazia Marinelli, Chiara Di Pede, Anna Maria Ciocia, Antonio Mancini, Giulia Palmieri, Giuseppe Barile, Vito Settanni, Nicole De Leonardis, Biagio Rapone, Fabio Piras, Fabio Viapiano, Filippo Cardarelli, Ludovica Nucci, Ioana Roxana Bordea, Antonio Scarano, Felice Lorusso, Andrea Palermo, Stefania Costa, Gianluca Martino Tartaglia, Alberto Corriero, Nicola Brienza, Daniela Di Venere, Francesco Inchingolo, Gianna Dipalma

**Affiliations:** 1Department of Interdisciplinary Medicine, University of Bari “Aldo Moro”, 70124 Bari, Italy; ad.inchingolo@libero.it (A.D.I.); giuseppinamalcangi@libero.it (G.M.); dralexandrasemjonova@libero.it (A.S.); angeloinchingolo@gmail.com (A.M.I.); assuntapatano@gmail.com (A.P.); giovanni.coloccia@gmail.com (G.C.); s.ceci@studenti.uniba.it (S.C.); graziamarinelli@live.it (G.M.); c.dipede1@studenti.uniba.it (C.D.P.); anna.ciocia1@gmail.com (A.M.C.); dr.antonio.mancini@gmail.com (A.M.); giuliapalmieri13@gmail.com (G.P.); g.barile93@hotmail.it (G.B.); v.settanni@libero.it (V.S.); nicoledeleonardis@outlook.it (N.D.L.); biagiorapone79@gmail.com (B.R.); dott.fabio.piras@gmail.com (F.P.); viapianofabio96@gmail.com (F.V.); drfilippocardarelli@libero.it (F.C.); daniela.divenere@uniba.it (D.D.V.); giannadipalma@tiscali.it (G.D.); 2Multidisciplinary Department of Medical-Surgical and Dental Specialties, University of Campania “Luigi Vanvitelli”, Via L. De Crecchio 6, 80138 Naples, Italy; ludovica.nucci@unicampania.it; 3Department of Oral Rehabilitation, Faculty of Dentistry, Iuliu Hațieganu University of Medicine and Pharmacy, 400012 Cluj-Napoca, Romania; 4Department of Innovative Technologies in Medicine and Dentistry, University of Chieti-Pescara, 66100 Chieti, Italy; ascarano@unich.it (A.S.); drlorussofelice@gmail.com (F.L.); 5Implant Dentistry College of Medicine and Dentistry Birmingham, University of Birmingham, Birmingham B46BN, UK; andrea.palermo2004@libero.it; 6Department of Biomedical and Dental Sciences and Morphofunctional Imaging, Section of Orthodontics, School of Dentistry, University of Messina, 98125 Messina, Italy; stefaniacosta94@gmail.com; 7UOC Maxillo-Facial Surgery and Dentistry, Department of Biomedical, Surgical and Dental Sciences, School of Dentistry, Fondazione IRCCS Ca’ Granda, Ospedale Maggiore Policlinico, University of Milan, 20100 Milan, Italy; gianluca.tartaglia@unimi.it; 8Department of Orthodontics, Faculty of Medicine, University of Milan, 20100 Milan, Italy; 9Unit of Anesthesia and Resuscitation, Department of Emergencies and Organ Transplantations, Aldo Moro University, 70124 Bari, Italy; alberto.corriero@gmail.com (A.C.); nicola.brienza@uniba.it (N.B.)

**Keywords:** dental caries, oral pathology, oralbiotica, oralbiotics, demineralization, remineralization, fluorine, orthodontics, probiotics, prebiotics

## Abstract

The oral microbiota plays a vital role in the human microbiome and oral health. Imbalances between microbes and their hosts can lead to oral and systemic disorders such as diabetes or cardiovascular disease. The purpose of this review is to investigate the literature evidence of oral microbiota dysbiosis on oral health and discuss current knowledge and emerging mechanisms governing oral polymicrobial synergy and dysbiosis; both have enhanced our understanding of pathogenic mechanisms and aided the design of innovative therapeutic approaches as ORALBIOTICA for oral diseases such as demineralization. PubMed, Web of Science, Google Scholar, Scopus, Cochrane Library, EMBEDDED, Dentistry & Oral Sciences Source via EBSCO, APA PsycINFO, APA PsyArticles, and DRUGS@FDA were searched for publications that matched our topic from January 2017 to 22 April 2022, with an English language constraint using the following Boolean keywords: (“microbio*” and “demineralization*”) AND (“oral microbiota” and “demineralization”). Twenty-two studies were included for qualitative analysis. As seen by the studies included in this review, the balance of the microbiota is unstable and influenced by oral hygiene, the presence of orthodontic devices in the oral cavity and poor eating habits that can modify its composition and behavior in both positive and negative ways, increasing the development of demineralization, caries processes, and periodontal disease. Under conditions of dysbiosis, favored by an acidic environment, the reproduction of specific bacterial strains increases, favoring cariogenic ones such as Bifidobacterium dentium, Bifidobacterium longum, and *S. mutans*, than *S. salivarius* and *A. viscosus*, and increasing of Firmicutes strains to the disadvantage of Bacteroidetes. Microbial balance can be restored by using probiotics and prebiotics to manage and treat oral diseases, as evidenced by mouthwashes or dietary modifications that can influence microbiota balance and prevent or slow disease progression.

## 1. Introduction

Half of the child population between the ages of 3 and 12 years develop tooth decay. The occurrence of caries is determined by multiple factors: dietary habits, composition and amount of saliva, fluoride intake, and imbalances in the oral microbiota (OM) [1]. Microbiota (MB) means the set of all microorganisms (bacteria, fungi, archaeobacteria, protozoa, and viruses) that, under precise physiological or pathological conditions and in a specific environment, live and colonize in symbiosis [2]. All the genetic heritage present in the MB and its expression is the microbiome (MM).

With metagenomics, the genomic sequencing study of 16S rRNA is carried out [3]. It is an RNA gene that produces the ribosomes responsible for protein synthesis. It is specific to each bacterium, by which the bacterial species is identified, its precise function understood, and thus, how the microorganisms interact with the specific site (microbial ecology) [4]. This is useful for evaluating diseases in which specific bacterial strains predominate [5,6]. Targeted technologies with transcriptomics (microarrays) and proteomics (electrophoresis, chromatography, mass spectrometry) respectively study gene expression by transcribed RNAs and all proteins synthesized from mRNA in an organism [7,8].

Metabolomics can also be added to these techniques, which tracks all metabolites produced by the microorganism involved in biological processes [9,10,11]. The human gastrointestinal tract, mouth, vagina, respiratory tract, and skin are colonized by microbial communities or microbiota [12].

The OM is heterogeneous and complex, comprising more than 700 families of microorganisms [13]. “Eubiosis” indicates the balance of the microbial system, “dysbiosis”, an imbalance with the proliferation of bacteria that tend to be pathogenic. Oral dysbiosis is a risk to oral health (tooth decay and periodontal disease) and the appearance of systemic diseases such as diabetes or cardiovascular disease [14,15]. About 75% of the OM manage to cross the gastrointestinal barrier to stabilize the gut microbial community in healthy people, even though distinct microbial communities are specific in each environment [14,16]. Many oral bacteria are found in the gastrointestinal tract of patients with colorectal cancer and rheumatoid arthritis [13,17,18,19]. Dental caries (DC) are caused by the increase of cariogenic bacteria, including Streptococci Mutans (MS), Lactobacilli, and several Actinomyces with a dysbiosis microbiome. With the metabolism of fermentable carbohydrates, these bacteria produce acid that demineralizes tooth surfaces [20,21,22]. Although DC is a multifactorial disease, the dysbiosis of the microbiome associated with dietary habits and sugar consumption could lead to its onset [16,23]. Studies have shown that the variety and combination of the OM in patients with metabolic diseases, such as diabetes, can change the composition and worsen teeth and gum health [24,25]. Saliva samples of adolescents with a mean age between 10.00 ± 0.67 years were analyzed, assessing glucose concentration, amount of bacterial plaque, and the number of bacterial species present among 42 strains examined (Table 1) [26]. Seven bacterial species (such as *N. mucosa*, *E. corrodens*, *S. mitis*, *P. melaninogenica*, *V. parvula*, *S. oralis*, *and S. salivary* were found that account for 50% of the bacteria analyzed in the salivary sample [24]. The body mass index and blood glucose levels were considered. A saliva glucose value < 1 mg/dL corresponded to normal blood glucose values (<100 mg/dL) and greater microbial diversity [27]. In contrast, in a situation of hyperglycemia (>100 mg/dL), saliva glucose concentrations were greater than 1 mg/dL, which were associated with a higher risk of DC and gingivitis, and lower bacterial load with a different composition of the oral microbiome [26,28]. Glucose is one of the major energy sources of bacteria; alteration of the microbiome in the saliva is a close consequence. The reduction of the bacterial variety in OM of patients with hyperglycemia would be due to the acidification of saliva [26,28] (Figure 1).

The studies demonstrate that when the blood glucose is above 84.8 mg/dL, it passes through the salivary glands from plasma to saliva [29,30]. Acidogenic bacteria thus increase the production of acidic metabolites, reducing the pH of saliva, which in turn interferes with the reproduction of specific bacterial strains favoring cariogenic ones such as *Bifidobacterium dentium*, *Bifidobacterium longum*, and *S. mutans*, that are more resistant to acids than *S. salivarius*, and *A. viscosus* [29,31,32].

At pH 4.2–4.4, in less than an hour, about 50% of the populations of *S. oralis* and Actinomyces are eliminated [33]. In obese patients, the oral microbiome prefers Firmicutes strains to the disadvantage of *Bacteroidetes* [34,35] (Figure 2).

Due to saliva acidification, demineralizing processes occur at the dental level and in the presence of gingivitis and periodontitis [36,37]. Furthermore, since 75% of gastric bacteria are derived from the OM, under acidic conditions, an acidic pH, the gastrointestinal barrier may inhibit the passage of gastric bacteroidetes, facilitating that of the gastric *Firmicutes* species [36,38,39,40].

From these studies, hyperglycemia, always present in obese patients and those with type 2 diabetes, increases the risk of demineralization, caries, and gingivitis. Nowadays, a large percentage of the pediatric and adolescent population performs orthognathodontic therapies. During orthodontic treatments, correlations of demineralization, DC, and gingivitis have occurred, primarily related to increased difficulty in performing careful hygiene with brackets and arches [41,42]. A difference in caries’ severity was noted comparing pcs with fixed therapy (FT) and pcs with aligner therapy (AT) [43]. FT results in more carious lesions and greater severity, although less extensive; AT, on the other hand, develops less deep, more extensive, and white-spotted lesions [44], therefore, using AT produces better expectations of oral health conditions [41]. A difference in higher plaque index was found between fixed lingual therapy and vestibular therapy. There was an increase of *S. mutans* in saliva samples in lingual therapy, with no major difference between *S. mutans* and *Lactobacillus* concentrations in patients treated with lip braces [45]. However, a good level of hygiene is maintained with elastomeric orthodontic appliances [46]. To date, fluoroprophylaxis is still essential in preventing caries [47]. The action of fluoride as prophylaxis against cariogenic processes occurs because fluoride transforms hydroxyapatite into fluorapatite, a more stable, less soluble, and more resistant compound [47,48,49,50].

It has bacteriostatic action, preventing the formation of the enamel cuticle and thus the adhesiveness of microorganisms to the enamel [51]. It inhibits bacterial growth by interfering with the permeability of the cell membrane [29,31,32]. Fluoride can be administered by: topic route, most effective, with high concentrations, is not to be ingested (gels, toothpaste, varnishes); or by Systemic route (per os drops, fluoridated waters, milk, salt, tablets) in individuals at high risk of caries [52]. In recent years, oral health has been determined by the variety of the OM and the absence of dysbiosis with the administration of pre and probiotics. Host microorganisms use prebiotics as substrates conferring an overall health benefit [53,54]. Urea and arginine are prebiotics for oral health and tooth decay [55]. They are produced by some oral bacteria, producing ammonia and raising pH. Although there is no guarantee chewing gum or mouthwashes with urea prevent the effect of caries, arginine added to foods has a real effect on demineralization [56,57]. Probiotics are microorganisms that improve host health and are administrated in balanced quantities. The use of milk and beverages enriched with probiotics and probiotic tablets can improve oral health in preschool children and those receptive to caries [58,59,60]. Recently, a study has shown effective rebalancing of oral microbiota using probiotics. They help reduce *S. mutans* in plaque and saliva. In combination with fluoride, their administration prevented or slowed the early demineralization of enamel [56,61,62]. In one study, a significant reduction in *Aggregatibacter Actinomycetemcomitans*, but not other periodontal pathogenic bacteria, occurred with probiotic administration in the oral cavity [63]. The science of probiotics using non-pathogenic oral bacteria, such as *Streptococcus Salivarius M18*, creates strain-specific and tissue-mediated antagonism by releasing bacteriocins (salivaricins) with antibiotic action [64,65]. A literature review showed that using probiotics regularly reduced the risk of caries with inhibitory action on cariogenic bacteria by enhancing the concentration of commensal bacteria in the oral cavity [66,67]. The potential of the probiotic on the attendance and endurance of *S. salivarius* and its ability to produce bacteriocins appears to be concentration-dependent, which is dose-dependent [64]. The *S. salivarius* would produce bacteriocins by gene sequences located in a megaplasmid, which would become repositories for bacteriocin determinants [68,69]. Streptococcus salivarius is present in greater amounts on the tongue’s surface. The release of bacteriocins by some bacterial strains would eliminate pathogenic bacteria [66,70,71,72]. Molecular studies have shown that these megaplasmids can synthesize molecules promoting host cell adhesion without generating antibiotic resistance [73]. To date, further studies on the influence of probiotics on the oral microbiota need to be further investigated [74]. The beneficial effects of probiotics are probably due to the ability to reduce salivary pH acidity, the production of bacteriocins and enzymes (dextranase, mutanase, and urease), inhibition of bacterial adhesion and colonization on dental areas, and possible immune system enhancement [64]. Studies are currently underway on the long-term benefits of probiotic supplementation on dental disease control and caries risk in childhood [75,76].

## 2. Materials and Methods

### 2.1. Search Processing

The current systematic review was carried out in compliance with the standards of the PRISMA and the International Prospective Register of Systematic Review Registry guidelines (full ID: CRD42022331431) [77]. PubMed, Web of Science, Google Scholar, Scopus, Cochrane Library, EMBEDDED, Dentistry & Oral Sciences Source via EBSCO, APA PsycINFO, APA PsyArticles, DRUGS@FDA were searched for publications that matched our topic from January 2017 to 22 April 2022, with an English language constraint. The search technique was developed by combining phrases that fit the goal of our research, which is primarily concerned with the influence of microbiota on tooth demineralization.; hence the following Boolean keywords (Table 1) were used: (“microbio*” and “demineralization*”) AND (“oral microbiota” and “demineralization”).

### 2.2. Inclusion Criteria

Reviewers have worked in duplicate and evaluated all suitable trials with the following inclusion criteria: (1) studies only on human subjects with dental demineralization. (2) open-access studies that other researchers can retrieve without any subscription. (3) studies that analyzed the link between dysbiosis of oral microbiota and the effects on the demineralization of the teeth with a particular focus on remineralization of the teeth with fluoride and the use of probiotics to rebalance microbiota; studies that did not take into account the oral microbiota were excluded.

### 2.3. Data Processing

Two independent reviewers (F.P., A.C.) assessed the quality of the included studies according to pre-defined criteria, including criteria for selection, methods of outcome assessment, and data analysis. Also, the quality criteria concerned in this modified ‘Risk of Bias’ Tool included selection, performance, detection, reporting, and other bias. Full texts were retrieved for any potentially relevant studies and then were identified according to the inclusion criteria. Any disagreements were resolved by discussion or consulting with a third researcher (F.I.).

## 3. Results

### Characteristics of Included Articles

A total of 1232 articles were found using ten databases, including Web of Science (156), Google Scholar (732), Scopus (63), Cochrane Library (1), EMBEDDED (0), Dentistry & Oral Sciences Source via EBSCO (47), APA PsycINFO (0), APA PsyArticles (0), DRUGS@FDA (0) which led to 1008 articles after removing duplicates (202), and non-English records (22). Searching the reference list of eligible articles produced 25 more relevant studies. The study of the title and abstract resulted in the exclusion of 977 articles. The remaining 31 reports were successfully retrieved and added to 1 paper discovered via reference list, resulting in 32 reports that the authors evaluated for eligibility. A total of 6 publications were rejected from discussion because they were off-topic. The evaluation included a total of 22 studies for qualitative analysis (Figure 3).

## 4. Discussion

### 4.1. Microbiota and Orthodontics

Tooth decay is a bacterial disease characterized by the localized and increasing destruction of the hard tooth tissues and cavity formation. Usually, the decay destroys the good part of the teeth’s crown and the infectious process of the pulp and peri-apical tissues. This infectious disease affects enamel, dentin, and cement if exposed to an oral environment, as in root decay. Tooth caries is one of the fifth most prevalent diseases globally; the disease is widespread in increasing numbers and is one of the most urgent problems for public health and the World Health Organization. The scientific society, the community, and healthcare professionals have focused on studying the etiology of this bacterial disease and its prevention. Critical preventive actions undertaken were dietary counseling, teaching oral hygiene maneuvers, and supplementing fluoride in water [42,78]. Tooth caries is a multifactorial disease [78] that originates from an alteration of the homeostatic balance between the native host and plaque microbiota [42,79,80,81]. In the cavity lesion’s genesis, this balance depends on the intake of carbohydrates in the diet. The role of the host is correlated to the enamel factor, saliva, and immune system [42,78,80,81]. In contrast, the role of the plaque is influenced by the bacterial composition, the capacity to produce organic acids, and diffusivity through the matrix. The existence of bacteria is an essential requirement for the genesis of the disease. The initial phase of tooth caries is the demineralization of the enamel, which lowers the pH [78,82]; this lowering is correlated to the organic acid produced by the bacterial plaque during the fermentation of different types of carbohydrates. Several studies (Table 2) studied the influence of orthodontic therapy on oral healthcare because introducing a foreign body represents a potential perturbation of the mouth system [81]; this adds to the risk present in every patient—the composition of saliva, an incorrect diet rich in sugar, and inadequate and insufficient oral health. We can add the development of new ecological niches provided by the fixed materials in the mouth. In addition, the patient presenting a malocclusion presents crowing that helps bacteria in the adhesion and colonization process of the surface through orthodontic appliance [42,78,80,81]. The initial settlement and adherence lead to the biofilm’s development, thanks to a complex interaction of species. It is a dynamic process that initially leads to changes in anaerobic conditions in the oral cavity sufficient to promote the spread of anaerobic bacteria, improving the state of illness [42,80]. 

Several articles studied the biofilm proliferation on the oral surface when present in an orthodontic appliance and are founded on the study of the saliva arrangement, especially in the bacterial characterization [42,78,80,81]. Bacterial plaques represent an adaptive microbial community, and species composition can change qualitatively and quantitatively. Initially, the process starts with the raising of plaque quantity. With the salivary sample collection, all articles studied the increasing amount of plaque, which depends on the patient’s difficulty in controlling oral hygiene [42,79,80,81,83]. Brackets and arches represent the metal surface that helps the enhancement of niches within the adjacent tooth surface. The material results have an important role, with the metal, cement, and composite used to fix the orthodontic device bringing changes in roughness and extension of roughness and attack surface [80]. Metallic brackets produced with stainless steel show the highest critical surface tension for the adhesion system [82]. Reichardt et al. [80] studied the quantitative modifications in the saliva, resulting in differentiation between the anterior and posterior regions because we have bands and cement in the posterior region while in the anterior region brackets and composite [82]. The study revealed higher plaque in the molar and premolar region because of metallic material and the greater device extension that increase plaque development [42,80]. Plaque development is a dynamic process that occurs when the orthodontic device is fixed and remains active at the end of treatment [42,80]. The bacterial change is not only quantitative but also qualitative. Reichardt et al. found differences in species [80]. At the beginning of the perturbation event, when the appliance was fixed, there was an increasing growth of the *Streptococcus Mutans* population, even thanks to the low patient hygiene. Bacterial species did not increase proportionally—some species grow faster than others, raising the risk of implementing caries [42,80].

Moreover, bacterial communities in anterior and posterior plaque were very similar [42,80], indicating that the microbial population is very stable during orthodontic therapy. Gujar et al. [82] pay attention to the periodontal species, finding a tendency to collect plaque when using a fixed lingual appliance more than the fixed labial appliance because of the simpler plaque accumulating on the lingual side, which is difficult to manage for the patient, leading to increase the subgingival pathogens. The aligner should be the orthodontic device that leads to a lower level of plaque accumulation [82]. Shokeen et al. [42] also compared fixed and not fixed orthodontic devices, concluding similarly that plaque’s holding was less when the appliance has no brackets, wires, bands, and other fixed metal products [42,80]. Plaque development represents an environmental stressor factor that changes the conditions of existence [42,78,80,81]. The bacteria in the mouth cavity can survive without changing in a wide range of environmental perturbations, such as variations in pH value, temperature, and oxidative stress [81,82]. Beyond the certain value of environmental perturbations, there is an increased susceptibility and higher risk of caries development in orthodontic patients. The introduction of local environmental stress, like fixed appliances, increases the *Streptococcus Mutans* population [81]. The introduction of orthodontic devices represents an environmental stressor and a foreign body that changes the patient’s oral habits. At the beginning of the orthodontic treatment, incorrect oral hygiene procedures can raise the risk of caries development [42,78,79,81]. The first step conducts the cavity formation is the clinical highlights of white spot lesion, the early sign of change in the enamel structure, visible at the clinical exam, and affects from 2 to 96–7% of orthodontic patients [44,78,79,81]. This incipient lesion is determined by the enamel demineralization and hydroxyapatite’s dissolution by the bacteria’s acidic product in the dental plaque [78]. Beerens et al. [79] studied this in incipient lesions, evaluating the amount of *Streptococcus Mutans* and *Lactobacilli* in the arrangement of saliva in the orthodontic subject. Although no variations in settlement of aciduric flora were found between patients with and without white spot lesions, we can confirm that the variable in the microbiological parameter is host-specific, and evaluating the bacterial composition is important to assess the individual risk of caries [79]. The increasing risk of decay in orthodontic patients can be limited by educating subjects to maintain constant and specific oral hygiene and oral health conditions. Another precaution is represented by using devices with fluoride to avoid the amount of *Streptococcus Mutans* species [83]. Padala et al. [83] conducted a clinical trial to demonstrate the relevance of prevention. Using elastomeric devices with a fluoride release shows that the proportion of Streptococcus Mutans decreases by about 30%, and this result is stable over time compared with conventional modules. Although fluoride released from the elastomeric device is localized, temporary, and needs continuing application, the results improve the role of prevention [44,78,83]. Likewise, Ali et al. [78] studied the prevention of using toothpaste containing fluoride to brush tooth surfaces because fluoride ions can prevent dental caries. The orthodontic device represents a stressor for the mouth cavity bacterial environment shift that changes the initial condition of existence. These articles highlight the impact of the increasing level of cariogenic bacteria, like *Streptococcus Mutans*, to enhance early enamel lesions, such as white spot lesions, initially and decay formation at last. Operators and patients must pay attention to oral hygiene and health during the orthodontic treatment, not compromise the mouth’s range of wellness, and not increase the cariogenic and periodontal complex.

**Table 2 children-09-01014-t002:** Included studies that explored microbiota and orthodontics.

Authors	Type of Study	Object	Study Design and Timeline	Results
Shokeen, 2021 [42]	Longitudinal study	Study bacterial shift in aligner orthodontic therapy	Microbiome analysis of supragingival plaque (16S rRNA) collected from 12 subjects at baseline and at 1, 3, 6, and 12-months after.	Orthodontic therapy is a stressor for oral environment. Clear aligner showed improved oral health condition
Padala RG, 2019 [83]	Randomized clinical trial, split-mouth	Impact of fluoride in elastomeric appliances to control Streptococcus mutans in orthodontic subjects	30 subjects, with 2 experimental periods of 3 weeks and a 3-week. Fluoridated elastics vs. non-fluoridated ones.	Fluoridated elastics were effective to decrease the Streptococcus mutans level in dental plaque.
Beerens MW, 2017 [79]	Cross-sectional study	Study caries risk assessment in orthodontic subjects	Dental plaque before debracketing and white spot lesions identified after debracketing. Microbiological analysis of the aciduric flora (*Strept. mutans*, *Lactobacillus and Candida a.)*	No significant differences between groups
Shimpo, 2022 [44]	Randomized control trial	Study the effectiveness of disinfection treatment, with the use of fluoride	White spot lesions identified by quantitative light-induced fluorescence, and bacteria by bacterial culture	Disinfection with PMTC and fluoride, shows efficacy in caries, when used on the tooth surface
Reichardt, 2019 [80]	Pilot study	Study qualitative and quantitative bacterial shift after insertion of orthodontic devices	Total 10 patients (12–15 years old). Microbiological samples at T0 and 1 week after at premolars and molars of the right upper side. Microbial evaluations by mass spectrometry	The orthodontic therapy leads to important shift in the oral bacteria, with gingivitis and raised risk of decay
Gujar, 2019 [82]	Clinical trial	Quantify orange and red bacteria in subjects with orthodontic aligners therapy, fixed labial, and lingual appliances	Total 60 patients, 20 treated with aligners, 20 with labial fixed appliances, and 20 with lingual fixed appliances. After a month brackets and aligners removed and DNA-DNA hybridization	After 1 month: bacterial contamination was major on brackets than aligners; in lingual fixed appliances was major than in labial-fixed appliances.
Ko-Adams, 2020 [81]	Clinical trial	In early orthodontic patients study the Streptococcus Mutans level, aerobic and facultative anaerobe bacterial PC quantitative	Plaque samples to evaluate the amount of Streptococcus Mutans and PCs from 17 patients after 1 month.	Statistically significant reduction in Streptococcus Mutans but not in PCs, and is highly variable across individuals
Ali, 022 [78]	Double-blind, randomized clinical trial study	Study the effectiveness of nano-silver, chlorhexidine, or fluoride mouthwashes on white spot lesions	Clinical examination of white spot lesions in 42 patients; 3 groups made to divide the type of brushing during the orthodontic therapy	Statistically significant discrepancies through the 3 groups. White spot lesions in the nanosilver group are lower than CHX and fluoride group

### 4.2. Microbiota and Remineralization

Sugars and their products allow the formation of a low pH habitat, which is favorable for the proliferation of bacteria that cause demineralization of the enamel. In the beginning, it is a reversible process that can be arrested and repaired with noninvasive preventive therapy [84]. Several studies (Table 3) have demonstrated how different substances support enamel remineralization by buffering the acidity of salivary ph. In the following section, studies concerning this topic were reviewed.

Yu-Rin Kim, Seoul-Hee Nam et al. investigated the bacterial composition of dental plaque and its pH using a mouthwash containing Glycyrrhiza Uralensis [85]. The study included 60 participants: 30 individuals gargled Glycyrrhiza Uralensis extract and 30 used saline gargle. Mouthwash was administered once a day. The researchers found that Glycyrrhiza Uralensis mouthwash could prevent dental caries and enhance dental health [90]. Bob T. Rosier, Carlos Palazon, et al. experimented with whether nitrate can help prevent mouth acidity caused by sugar fermentation. Twelve people ingested a nitrate-rich substitute diluted in mineral water (250 mg nitrate per 200 mL) and a nitrate-poor placebo diluted in mineral water [84]. Six people received the active supplement one day and the placebo the next week, while the remaining six did the opposite. This research demonstrates that nitrate could buffer pH values due to lactate reduction during sugar fermentation. This aspect is connected with bacterial reduction as Rothia and Neisseria. Anie Apriani et al. experimented with how fluoride varnish compares to casein phosphopeptides-amorphous calcium phosphate (CPP-ACP) regarding lowering saliva pH and caries activity [48]. Sixty patients were enrolled: 30 patients in each group. Group 1 received fluoride varnish, while Group 2 received CPP-ACP. Using the varnish for one month is the most effective treatment for raising saliva pH and lowering caries activity [47]. However, fluoride varnish and CPP-ACP have no statistically significant differences in changing saliva pH and lowering caries activity. Rahul G. Padala et al. analyzed the impact of fluoride-releasing elastic modules on Streptococcus mutans levels in the mouth [86]. The study had 30 orthodontic patients (14 men and 16 women) undergoing treatment. Modules with fluoride were located on brackets of 12-11-33; instead, non-fluoridated ones were adapted on 21-22-43, then proceeded with microbiological analysis. In a single visit, nonfluorinated modules were applied to all brackets. At the third visit, fluorinated elastic modules were adapted to brackets 21-22-43, and nonfluorinated elastic modules were put to brackets 12-11-33 [51]. In the fourth session, the procedures from the second visit were recreated. The Mann–Whitney U test was performed in both studies to evaluate the microbial activity of fluoride-releasing elastic modules, and the outcome was expressive (*p* 0.001). M.M. Nascimento et al. studied the metabolic profile of supragingival plaque after arginine or fluoride treatment [87]. There were 83 subjects. The International Caries Detection and Evaluation System II was used to examine plaque from carious and non-carious tissues. Human Oral Microbe Identification (HOMINGS) was used to access plaque metabolism. In particular, arginine catabolism, acidogenicity, and metabolomic and taxonomic profiles [91]. To identify metabolic patterns, researchers utilized principal component analysis (PCA), partial least squares–discriminant analysis, analysis of variance, and random forest tests [92]. The study found that fluoride is hypothesized to improve tooth mineral resilience to acidic pH and limit acid generation by biofilms, whereas arginine metabolism improves biofilm pH equilibrium. Research by Xin Zheng et al. verified that arginine could modulate in vitro microbiome, suggesting it could be a viable anti-caries drug. However, its impact needs further investigation in clinical cohorts that more accurately reflect the oral microbial variety [88]. Authors verified that oral microbiota can be normalized through arginine-based toothpaste. Furthermore, they discovered that arginine and fluoride together improved the enrichment of alkali-producing Streptococcus sanguinis while suppressing acidogenic/aciduric Streptococcus mutants, leading to a considerable loss in the demineralizing ability. Fluoride and arginine combined are synergic in promoting a healthy dental microbial balance and better caries management. Mine Koruyucu et al. research compares different properties of fluoride and no-fluoride toothpaste [89]. 80 patients (from 3 to 12 years old) were randomized and divided into four different groups and each mouth evaluated. The first two groups (40 patients aged 6 to 12) used fluoride toothpaste, while the last two groups (40 patients aged 3–5) used non-fluoride toothpaste. Results showed that both could reduce *S. mutans* presence, but no statistical differences were found between fluoride and no-fluoride toothpaste [86]. In his research, Azheen Ali evaluated the influence of nano-silver, chlorhexidine (CHX), and fluoride mouth rinses on white spot lesions (WSL). For this investigation, 42 individuals were enrolled in a double-blind prospective randomized clinical trial [78]. Block randomization was utilized to split the patients into three different groups: each group was subjected to different mouthwash therapy and evaluated at 90 and 180 days from the start of therapy. The nanosilver group is significantly lower associated with WSL compared to CHX and fluoride groups [93]. Many authors have analyzed exogenous substances that potentially alter the oral microbiome worldwide. Various studies carried out about saliva PH: Glycyrrhiza Uralensis mouthwash, a nitrate-rich supplement that contributes to reducing the acidity of salivary pH, while casein phosphopeptides-amorphous calcium phosphate and fluoride varnish don’t show any differences between both. Particular attention was conducted to the main responsible for dental caries: *S. mutans*. The literature describes that it is susceptible when treated with fluoride-releasing modules and fluoride-rich toothpaste; moreover, fluoride contributes to increasing the enamel resistance to acidic pH, and combined with arginine, can maintain oral microbial equilibrium.

With the advent of biomaterials and nanotechnologies, bio-reactive and biomimetic products represent a new approach to preventing erosion and dental caries. These products differ from bone and cannot be repaired when hydroxyapatite is damaged because the enamel has no cells [94]. These products’ remineralizing and repairing properties are due to the coating of the enamel surface by micro-particles that create a biomimetic film and reproduce the biologic hydroxyapatite. Indeed synthetic biomimetic carbonate hydroxyapatite nanocrystals are very similar to biogenic nanocrystals and produce a less crystalline layer than native carbonate apatite and fill enamel scratches and pits, as well as dentin-exposed tubules [94]. The treatment with biomimetic hydroxyapatite nanocrystals toothpaste may also reduce dentine hypersensitivity, as demonstrated by Orsini et al. in numerous clinical randomized trials [95,96]. This desensitizing effect of biomimetic materials is due to the closure of dentinal tubules by the deposition of this mineralized layer [97,98]. Finally, despite the significant impacts of biomimetic toothpaste, long-term investigations of its qualities are required.

### 4.3. Microbiota and Dental Demineralization

The body’s symbiotic microflora is biologically crucial for our organism’s whole life cycle. It controls several important physiological, biochemical, and immunological aspects. Microflora, the normal intestinal microbiota, has been shown to increase the generation of secretary immunoglobulin and maintain high mucin levels [99]. The physiological flora prevents harmful organisms from colonizing and reproducing. The multiplication of decomposing microbes inhibits the creation of lactic and succinic acid by *Bifid* and *coli bacteria* in the *GI. Bifidobacteria* are involved in the manufacture of iron, zinc, calcium, and vitamin D and the absorption of amino acids. They also produce B vitamins, folic acid, nicotinic, and pantothenic acids. As a result, changes in the GI microflora’s quality and quantity have a detrimental impact on the organism’s ability to fight infection and enhance cells’ allergenic and mutagenic potential [100]. Intestinal dysbiosis is a bacteriological imbalance constituting an imbalance in the proportion and type of the gastrointestinal microbiota. The lack of or full absence of essential microorganisms is the main cause of this process [101]. The degree of pathogenicity of microorganisms, the patient’s age, somatic disorders, the influence of adverse ecological and environmental elements, the use of antibacterial medications, or an improper diet all contribute to intestinal development dysbiosis [102]. The continuation of the GI tract and the oral cavity causes GI dysbiosis in children to manifest as oral alterations. *Staphylococci*, *streptococci*, *lactobacilli*, *fungi*, *corynebacteria*, and different anaerobes make up most oral and gastrointestinal microflora. According to the dysbiosis theory, the modern diet, lifestyle, and antibiotic usage have disrupted the typical gut microflora. These variables cause changes in bacterial metabolism and increase the number of potentially harmful germs. It is known that the presence of chemical elements is unfavorable for the bacteria activity in the intestine and determines a key factor in chronic and degenerative disorders. Antibiotic usage, psychological and physical stress, radiation, changed GI peristalsis, and dietary changes are just a few variables that can affect the beneficial members of the GI flora [103,104,105,106,107]. Bacteria have a high capacity for binding calcium ions of the oral cavity proteins, which has ramifications for various functions, including signaling, sugar and protein transport, aggregation, and biofilm generation [108,109,110]. This feature has significant relevance to the biofilm of tooth decay. The calcium and the hydroxyapatite ions can counteract and decrease the demineralization of the teeth when attacked by bacteria and with liquid biofilm [21,111,112]. Furthermore, calcium may link the anionic groups on the microorganism’s surface, and fluoride has an important consequence on dental caries, lowering demineralization and encouraging dental auto-remineralization [51,113]. A recent study by Domon-Tawaraya et al. showed how calcium from saliva combines with the topical fluoride of bacteria from the dental biofilm [114]. The study also notes how much calcium at high concentrations determines the development of precipitated minerals, including calcium fluoride. This happens so that the topical fluoride intervenes in the entire bacterial metabolism. Dentists must develop the best remineralizing procedure to consolidate the solidity of external hard tissues consisting of dentin and enamel in cases of intestinal dysbiosis treatment [115]. The present study (Table 4) [115] has highlighted how gastrointestinal dysbiosis has a specific influence on the mineralization of mineralized surfaces of the tooth, for which the worsening of dysbiosis determines the level of dental diseases [115].

Since next-generation sequencing methods became accessible, the microbe’s oral groups have been extensively explored, including organisms that are difficult to cultivate [121,122]. According to these studies, oral microbiota has a significant task in oral health [123,124]. Several illnesses, including caries and periodontitis, have been linked to beneficial eubiotic and harmful dysbiotic microbiotas [124]. There are, however, some oral disorders for which no bacterial link has been shown. The molar-incisor hypomineralization (MIH) is an example of this. Because hard dental tissue lacks a repair mechanism, mineralization or maturation problems of the outer tooth covering manifest in the tooth corresponding to the developmental stage [125]. Disorders during the initial matrix secretory stage of amelogenesis can cause quantitative structural defects, such as dental hypoplasia. Those that impact the maturation or mineralization stages can cause hypomineralization or qualitative problems [126]. During the European Academy of Paediatric Dentistry meeting in Athens in 2003, the term “molar-incisor hypomineralization” (MIH) proposed by Weerheijm [127] was accepted for defining a “hypomineralization of systemic origin and etiology that affects at least one permanent molars and it often involves permanent incisors” [128]. MIH manifests as asymmetric opacities of white, cream, yellow, or brown color in the cusp or incisal third of the crown of the affected teeth, with different degrees of extension and severity [129]. MIH usually appears during early childhood, when the crowns of the first permanent molars (FPM) and permanent incisors (PI) begin to mineralize [130,131]. MIH can cause major hypersensitivity and pain, post-eruptive breakdown, chewing and eating issues, esthetics, and therapeutic challenges. MIH is also difficult for patients, carers, and dentists due to poor restorative and therapeutic outcomes [132]. Although MIH has been linked to prenatal exposures to potential risk factors, the most reasonable explanation is a complex etiology with a possible genetic component [133]. The cause is related to several activities associated with the reabsorption of organic matrix and the cancellation of proteolytic enzymes. As a result, proteins would accumulate, decreasing the space available for mineral deposit [130], resulting in porous enamel [134]. In fact, MIH affects the thin outer covering of the tooth; it has from 3 to 21-fold higher protein concentration than healthy enamel, and contemporary decrease its hardness and flexibility [135]. 

When healthy and dysbiotic microbiotas are examined, the healthy samples usually have more diversity, as seen when comparing healthy teeth surfaces to enamel or dentin caries lesions [136]. In some cases, the opposite was found 3 in periodontal pockets compared to healthy elative space between each tooth and the gum tissue surrounding it [137]. Increased nutritional availability or a compromised immune system in the affected areas have both been connected to microbial colonization [124]. According to scientific research, porosity is regulated by the quality of the enamel, which would also impact the higher exposure of the attachable tooth surface [116]. Moreover, increasingly complex microbial colonies, large and especially proteolytic, are implemented by the greater presence of proteins in MIH lesions. It is significant to note that patients with softer teeth due to MIH disease tend to make oral hygiene more frequently and consequently determine an uneven distribution of plaque. Streptococcus (12%) and Leptotrichia (10%) were the most common genera in these two instances, whereas *Prevotella*, *Fusobacterium*, *Capnocytophaga*, *Selenomonas*, *Corynebacterium*, *Veillonella*, *Porphyromonas*, *Neisseria* and *Saccharimonadaceae* had similar percentages (2–5%), reaching 70%. Unclassified *Clostridiales Family XIII*, *Fusobacterium*, *Campylobacter*, *Tannerella*, *Centipeda*, *Selenomonas*, *Streptobacillus*, and *Alloprevotella* have all been linked to MIH patients (Figure 4) [116].

Furthermore, numerous research studies linked the Hypo-related species to periodontitis. Centipeda periodontii, for example, has been linked to periodontitis [138]. Siqueira and Roças detected *Catonella morbi* in 33% of root canals in chronic apical periodontitis and 26 percent of primary endodontic infections by amplifying the 16S rRNA gene [139]. MIH is linked to the advancement of caries due to lower inorganic components and higher exposure to infections and periodontal and bone diseases that surround and support the teeth due to increased amounts of bacteria. Furthermore, *Fusobacterium*, *Catonella*, *Campylobacter*, *Selenomonas*, *Alloprevotella*, and *Centipeda* are present in MIH disease and related with oral cancer [140]. Because the high quantity of bacteria discovered in hypomineralized lesions are proteolytic, it is possible that the higher protein content in MIH sites stimulates microbical development, which has been associated with various oral and structural disorders [116].

Concerning the context of pregnancy, a comparison of microbial abundance in the oral cavity of non-pregnant women versus pregnant at the beginning, middle, and end of pregnancy was made, revealing that counts are increased at all phases of pregnancy, with a focus on the early stages. Further research has indicated that infections such as Porphyromonas gingivalis and Aggregatibacter actinomycetemcomitans are more prevalent in pregnancy’s early and middle stages [141]. In contrast, other writers have discovered more significant amounts of *A. actinomycetemcomitans* even in the third quarter [142]. Candida levels were greater in the middle and late stages of pregnancy than in non-pregnant women. Some writers speculate that progesterone and estrogen levels may explain these differences in the pregnant woman’s oral microbiota. However, except for estrogen’s impact on Candida levels, the remainder of the hypothesis has yet to be rigorously verified [141,143]. Hard tissue is required to colonize cariogenic bacteria, notably *S. mutans*, in the oral cavity. Bacterial species are typically transmitted vertically from mother to kid between the ages of 19 and 31 months. It has been demonstrated that delaying the colonization of bacteria in the child’s oral cavity reduces the likelihood of acquiring carious lesions at seven years.

In contrast, early colonization corresponds with early childhood caries (ECC) and MIH [144,145]. Colonization patterns in the oral cavity differ between three-month-old infants born vaginally and those delivered through Caesarean section. Possible explanations for discrepancies include the differential importance of host receptor and mucosal and saliva immune phenotypes and interactions with environmental exposures [146]. With probes to the 16S rRNA gene of cultured and uncultivated oral bacteria in a microarray format, more taxa were found in vaginally delivered newborns than in C-section babies [147]. Furthermore, the data indicated microbiome differences dependent on delivery strategy, including the unusual observation that Slackia exigua was exclusively in high prevalence in newborns delivered through C-section. These data suggest that the microbiota of the mouth cavity differed depending on the delivery type, as has been observed for the bacteria of the lower gastrointestinal tract [148,149].

The microorganism of the human mouth plays a vital role in health by preventing pathogenic species from colonizing the mouth cavity [150]. Many bacteria habit the human oral cavity, with meta-genomic studies identifying more than 700 species [151]. The mouth cavity has various biological niches that are influenced by nutrient supply, pH, and oxygen tension. Subgingival biofilms, for example, have characteristics that differ from biofilms above the gingival border or on the tongue. To examine if there were any differences in acid tolerance between the supragingival regions of the patients, researchers looked at the biofilm of the incisor, premolar, and molar surfaces. The acid tolerance of the individual is quite uniform throughout all three locations, particularly between the incisor and molar areas. This shows that supragingival biofilms on various teeth are exposed to the same stressors and ecological cues important for developing acid tolerance [150]. Bacteria form in multi-species biofilms on hard and soft oral tissue surfaces, with saliva or gingival discharge as the primary nutrition supply. Sacharolytic bacteria dominate biofilms on teeth’s dental plaque above the gingival edge, which creates energy by breaking down carbohydrates from salivary glycoproteins and ingested food via the glycolytic route. The acidic end-products of pyruvate conversion, such as lactic acid, can quickly reduce the pH of dental biofilms. Demineralization of the enamel occurs when the pH falls below 5.5 over an extended time [152,153]. Acids produced by members of the biofilm’s carbohydrate metabolism disrupt the biofilm’s ecology and cause demineralization. Microorganisms metabolize carbohydrates from meals periodically, producing chemicals such as acetate, lactate, formate, and succinate, which cause rapid acidification of the biofilm [152]. Stephan [154] showed in 1944 that after a glucose rinse, the pH of plaque from caries patients at the beginning was lower and remained the same for longer than plaque from healthy participants. Long periods of low pH in biofilms encourage the growth of bacteria that are naturally acid-tolerant (aciduric), such as *Lactobacilli* and *Bifidobacteria* (which have adhesins that may bind adsorbed salivary proteins), resulting in an increased number of these bacteria [155]. Experiments with the model bacteria Streptococcus mutans have demonstrated that, although not innately aciduric, oral streptococci can resist acid stress by inducing an acid-tolerance response (ATR) when subjected to a pH of less than 5 [156,157]. The ability of a bacteria to sense and adapt to acid stress is known as acid tolerance, and it was initially observed in Salmonella enterica serovar typhimurium [158]. Later, it was discovered in Escherichia coli, Listeria monocytogenes, Bifidobacterium longum, and oral bacteria such Enterococcus Hirae, Streptococcus Gordonii, Streptococcus sanguinis, and *Lactobacillus* Casei [157,159,160,161,162]. One of the most common chronic infectious illnesses is damaged hard tooth surfaces [163], characterized by the destruction of hard and mineralized dental tissues as a consequence of a variety of factors, including bacterial metabolism, a high-sugar diet, poor oral hygiene, and the patient’s intrinsic susceptibility and socio-cultural elements (i.e., level of education and employment status) [164]. Acid compounds produced by commensal bacteria in the oral biofilm [165] cause demineralization of tooth areas and caries. “Specific Theory” and “Ecological Plaque Theory” is based on the suppose that the incidence of caries is correlated to the presence of specific bacteria such as Streptococcus mutans and Streptococcus Sobrinus [166], resulting in commensal microbiota dysbiosis [167,168]. Different clinical conditions are associated with dysbiosis and the selection mechanisms of specific microbial species [168,169]. Biofilm development is a manifestation of pathogenicity in microorganisms [170]. Polymicrobial biofilms developed on the tooth and mucosal surfaces and implants and tooth materials can determine a lot of diseases and difficulties in the oral cavity [171]. Oral biofilm dysbiosis influences the proliferation of acidogenic cariogenic bacteria like Streptococci, Lactobacilli, and Candida [168,169]. Patients consider traditional caries removal methods, like manual and mechanical rotating devices (dentine spoon, turbine, or drill), painful. The pain can be alleviated with a local anesthetic while discomfort from noise, vibration, or dread of the needle persists. Mechanical procedures may injure the pulp due to thermal stimulation and remove much healthy tissue. Different alternative therapies have lately been presented to protect the tooth structure by eliminating only the affected tissue and being as non-invasive as possible [172,173,174]. Although ozone therapy provides many benefits in dentistry, the dangers are still too great due to upper airway difficulties and contraindications in fragile patients [175]. Lasers, in particular, have a dual effect (mechanical and thermal) and can disorganize microbial biofilms. This is significant because if pathogenic microorganisms in the cavity are not properly eradicated, the carious lesion might grow and evolve, damaging the pulp [176]. In conservative dental procedures, the leftover microbes might also be a source of recurrent caries. The use of laser therapy affects bacterial adhesion to tooth surfaces and microbial load decrease [177,178]. Furthermore, several oral environment elements connect with microbial communities [179,180]. Intraoral pH, for example, has been shown to significantly impact the organization of microbial communities, particularly in partial dentin caries-associated bacteria such *Lactobacillus* species [92,181,182]. Salivary iron, an important elemental metal in saliva, has also been demonstrated to alter the salivary microbial profile and provide essential nutrition for oral bacterial species [183,184]. The bacterial community structure was also substantially linked with the DMFT index, a measure of dental caries load. A higher DMFT score was consistently linked to a lower pH and higher iron content [119]. These findings suggest intraoral pH monitoring could assess lesion acidity in the clinic [92], although they contradict iron’s cariostatic properties [183]. The disintegration of the dental structure by acid generated by oral microorganisms due to the fermentation of dietary carbohydrates is well understood in the genesis of dental lesions, with aciduric or acid-producing species being the most frequent common cariogenic species [185]. A certain iron level could prevent enamel demineralization by forming an acid-resistant layer on the thin upper dental surface [183,186,187,188,189,190,191].

### 4.4. Microbiota and the Influence of Probiotics and Prebiotics on Dental Remineralization

Probiotics are beneficial microorganisms that help maintain the digestive tract’s bacterial equilibrium [61,70] (Table 5).

Probiotics must first aggregate and bind to the oral tissue to build a protective barrier that inhibits harmful microbe invasion [70,192]. Substances containing live bacteria sustain the adhesion to mouth surfaces, colonize them, and battle for three the binding site with harmful microorganisms [193]. Prebiotics are nondigestible oligosaccharides utilized to activate helpful microorganisms selectively [192,194]. They improve probiotics’ therapeutic benefits by increasing their growth and activity [195]. Humans have a favorable relationship with their mouth microbiomes, which is essential for mouth relationship management [196]. Acidogenic microbes have benefited from a low pH environment caused by regular carbohydrate consumption in the mouth cavity. This altered environment stimulates the activity and proliferation of acidogenic and aciduric bacteria, including *S. mutans* and *Lactobacillus* spp. Only 2% of *S. mutans* is identified in the dental biofilm under equilibrium [70]. The proportion of *S. mutans* and Lactobacillus spp. Increases significantly when the balanced environment is changed to anacidic [197]. Caries of the teeth are a serious oral disease that affects children and adults. It has been noted that the bacterium with the highest cariogenic factor is *S. mutans* which exploits its acidogenic and acidulous structure to metabolize sugars. Many studies have confirmed that the addition of probiotics, such as *Lactobacillus* acidophilus, changes the harmful extent of *S. mutans* [70]. Therefore, a step forward has been made in exploiting probiotics’ potential, raising a protective firewall against pathogenic microorganisms, and supporting the host’s self-defense [198]. For years, many studies have converged to demonstrate some new therapy guidelines that introduce the use of probiotics to counteract mouth disorders [199]. The combined action of viruses and bacteria [200] proves to be the most important cause of oral and respiratory diseases [61]. Furthermore, combined adverse action determines immune dysfunctions, cyclic respiratory tract infections (RTI) with important bacterial infections, worsening cough, and a higher incidence of asthma [61,201]. Streptococcus pneumoniae, Mycoplasma pneumoniae, and Streptococcus pyogenes [202] are the most common bacterial respiratory pathogens. One severe issue with such illnesses is the patient’s age; in reality, most children contract them when under the age of two, and 25% of them in industrialized countries suffer from repeated or extended infections [203,204]. Because RTIs affect fragile children, these pathological forms impede physical and scholastic activity; in fact, RTIs are a leading cause of school absenteeism and hospitalization [205,206]. The reasons for the essential medical check-ups or the otolaryngologist are the pathologies of the upper respiratory tract. It has been found that among the most crucial cause of the pathologies mentioned above are bacterial superinfections developed from the excess use of recommended antibiotics with a frequency of 60–80% of cases [207,208,209]. Antibiotic usage may lead to bacterial resistance and disturb the normal microbiota balance, favoring pathogen colonization and decreasing viral vaccine accessibility [210,211]. Germs are crucial in protecting their environment from pathogens that could damage overall health [61]. Metchnikoff developed probiotics over a century ago, believing that using specific bacteria could assist in maintaining health. Various mouth disorders are caused by alterations in oral and respiratory bacterial flora, and the idea of using probiotics is paving the way for new treatment perspectives [199]. Many human diseases cause a change in the ideal condition of microbiota. The most frequent presence in the oral cavity is *S. salivarius*, a common commensal of the oral cavity. *S. salivarius* is a rich source of effective probiotics capable of implementing the most proportionate and significantly associated oral microbiota with oral health [212]. One of the most common bacterial factors causing pharyngeal infections in humans is Streptococcus pyogenes inhibited by a specific probiotic, *S. salivarius* K12 (SSK12). Multiple updated studies linked daily SSK12 usage to a considerable reduction in streptococcal pharyngitis and acute otitis media, particularly in children. We established that probiotics had a tangible link with dental and respiratory health [213]. Probiotics have been shown to have local effects on the microbiota of the digestive system, such as lowering possible infections, modifying gut functionality and permeability, and acting as immunomodulatory drugs in patients. Probiotic exopolysaccharides can influence both innate and adaptive immunity [214]. Substances containing live bacteria can also increase antibacterial action against oral infections by increasing salivary immunoglobulin A levels [215,216]. A variety of severe intestinal mucosae disorders have been linked to probiotic microorganisms. Notably, probiotics have recently been linked to reducing chronic constipation by significantly decreasing methane production [217]. The most excellent intriguing findings concerned the advantages of probiotics and gut microbiota on rotavirus-related gastrointestinal severe diseases prevention and therapy. Several studies have been conducted to investigate the influence of probiotics on the oral and respiratory tracts [61]. Santagati et al. [218] reported that *S. salivarius* lacks any virulence factors but is a powerful generator of bactericidins against *S. Pneumoniae*, the most prevalent respiratory bacterial infection. Multiple cytokines linked with inflammatory ear-nose-throat diseases were effectively suppressed by *Lactobacillus* living on the respiratory mucosa [219].

## 5. Conclusions

The study of the oral microbiota represents a flourishing and interesting field of research to understand the links between oral health and the health of the whole organism. Advances in “omics” technologies (genomics, transcriptomics, proteomics, and metabolomics) have enabled and will enable an understanding of oral microbiota’s physiology and pathogenetic role. The bacteria, fungi, and viruses that inhabit the oral cavity cannot be considered independent entities. However, they must be studied as a whole to understand the networks and interactions among microorganisms that operate in concert in the onset of disease or maintenance of a healthy state. The oral ecosystem consists of a complex and heterogeneous microbiota comprising more than 700 families of microorganisms that coexist in balance and equilibrium with the host organism. Disruption of this balance is called “dysbiosis”. It can be related to the development of oral diseases such as demineralization, caries, and periodontal disease, but it also interacts with systemic diseases such as diabetes and cardiovascular disease. As can be seen from the articles included in this review, the microbiota balance is fragile and affected by multiple factors that can positively and negatively alter its composition and behavior. Oral hygiene, the presence of orthodontic devices in the oral cavity and improper eating habits are all factors that can alter the composition of the oral microbiota by promoting the onset of areas of demineralization, caries processes, and periodontal disease. “Oralbiotica” (oral: a procedure performed on the teeth, gums, jaws, or other oral structures; bio: use of viable bacterial forms; tica: specific controlled application) defines a specific approach to the daily interaction between bacteria of the oral cavity by utilizing their strengths and adopting a real-world therapy known as “bacteria control”, realizing a new type of interaction. This method is based on the ability to establish competition between bacterial species in pathological tissues of the mouth cavity. Therefore, it is not just using typed strains with fully decoded genotype and phenotype. In order to precisely and efficiently execute the new “oralbiotic” method, the dentist has to conduct a complete evaluation of the patient and prepare the use of fluoroprophylaxis, prophylaxis, and remineralizes to achieve an allopathic result. The “oralbiotica” implemented by the dentist creates a physiological habitat for eubiotic bacterial species.

Moreover, biotics favorably affects the oral microbiota by reducing the charge of bacteria responsible for stomatological pathologies e it represents an excellent multi-therapeutic method that can reduce the incidence of relapses by defeating infections with the same infection-fighting weapons used in eastern self-defense. Oralbiotica can also be seen as a new approach in preventive dental care that emphasizes the importance of ongoing oral procedures and daily practices to prevent tooth decay and other dental diseases and conditions.

According to the research, certain substances actively promote the mineralization of dental enamel by modifying the acidity of the salivary pH. Glycyrrhiza Uralensis mouthwash, nitrates, and fluoride varnish, for example, can prevent the formation of dental caries and improve oral health.

Furthermore, fluoride and arginine have shown an essential balancing effect on the oral microbiota, playing a primary role in antagonizing the main responsible for dental caries: *S. mutans*. Effective preventive dentistry combines chairside treatments, counseling by dental professionals, and at-home oral care by patients.

Knowing the mechanisms through which microorganisms interact with each other allows opening a new frontier in the treatment of these pathologies: seeking the balance of the microbiota using probiotics and prebiotics in order to control and treat oral pathologies, as already demonstrated through the use of mouthwashes or dietary implementations capable of modifying the balance of the microbiota and halting or reducing the progression of pathologies.

The main limitation to the interpretability and generalizability of these results is the dissimilarity of the studies chosen for inclusion in our review. The revisor did not undertake a meta-analysis of the results of all studies published due to the heterogeneity of the studies reviewed. The differences in materials and methods, as well as the specific goals of the various studies, are potential sources of bias and heterogeneity.

Studies with larger sample sizes analyzing different health and disease conditions are required to develop consistent models and generate more solid data. Standardization of scientific methodology is critical to handling the data sets that will be generated. In future studies, standardization and conceptualizing clear hypotheses are essential to achieving high-quality and valid outcomes.

A future goal of scientific research should be to produce scientific trials with consistent samples, as well as meta-analyses that can unify and clarify the results of individual studies and investigate the correlation between microbiota and multiple diseases, oral and systemic, in order to shed light on the complex molecular pathways involved in pathogenic processes. Furthermore, prospective longitudinal studies are required to reveal the full potential of the oral microbiome in personalized medicine and identify new biomarkers to develop increasingly targeted therapies.

## Figures and Tables

**Figure 1 children-09-01014-f001:**
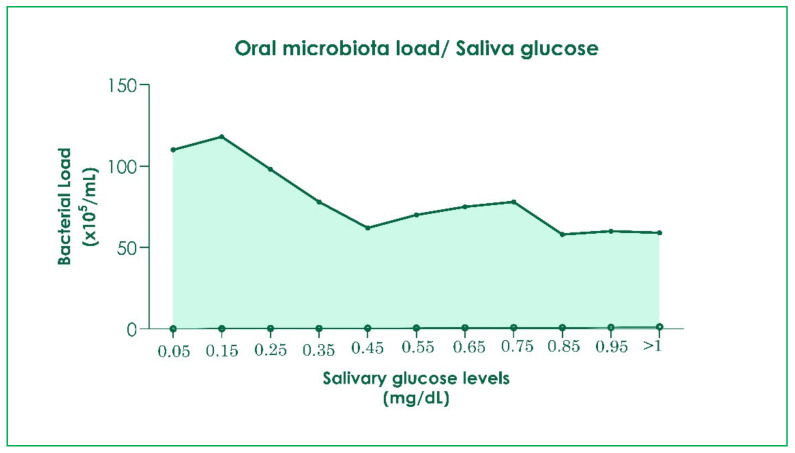
Deflection curve of OM load with an increase of glucose in saliva [29,30].

**Figure 2 children-09-01014-f002:**
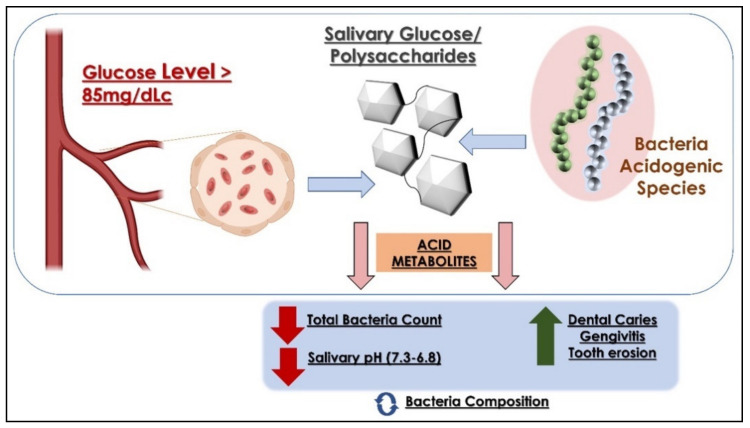
The proposed hypothesis to explain changes at the level of the OM with a high level of glucose in saliva.

**Figure 3 children-09-01014-f003:**
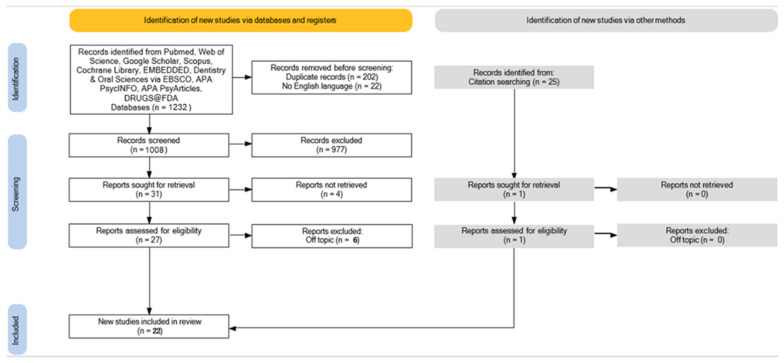
PRISMA flowchart diagram of the inclusion process.

**Figure 4 children-09-01014-f004:**
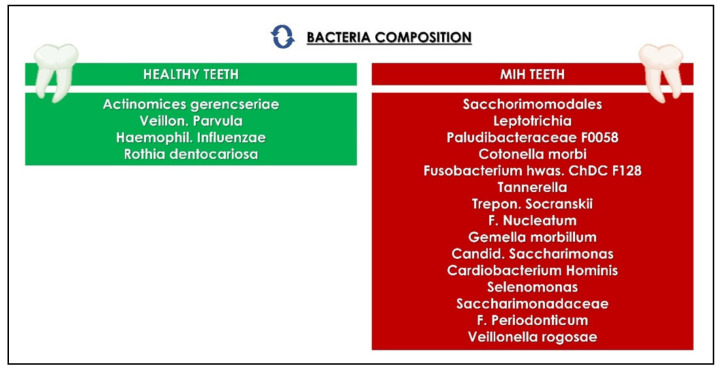
Biological differences between teeth affected by MIH and undamaged teeth [116].

**Table 1 children-09-01014-t001:** Database search indicators.

Articles screening strategy	KEYWORDS: A: “microbio*”; B: “oral microbiota”; C: “demineralization”; Boolean Indicators: (“A” AND “C”) AND (“B” AND “C”). Timespan: from January 2017 up to April 2022. Electronic Databases: PubMed, Web of Science, Google Scholar, Scopus, Cochrane Library, EMBEDDED, Dentistry & Oral Sciences Source via EBSCO, APA PsycINFO, APA PsyArticles, DRUGS@FDA

**Table 3 children-09-01014-t003:** Included studies that examined remineralization. [CPP-ACP: Casein phosphopeptides-amorphous calcium phosphate; CHX: Chlorexidine; WLSs: white spot lesions].

Authors	Type of Study	Object	Study Design and Timeline	Results
Yu-Rin Kim et al. [85]	A Randomized, Double-Blind, Placebo-Controlled Clinical Trial	Dental plaque pH variations after Glycyrrhiza Uralensis mouthwash.	Total of 60 patients, (30 case group and 30 control group), treated with placebo mouthwash. Mouthwash administered one time a day/5 days.	Glycyrrhiza Uralensis mouthwash is useful to prevent dental caries.
Bob T. et al. [84]	Blinded crossover study	Effect of nitrate on reduction of oral pH by sugar fermentation	Total 12 subjects subjected to nitrate-rich supplement dissolved in mineral water and a nitrate-poor placebo dissolved in mineral water vs. placebo.	pH buffering effect of nitrate when sugars were fermented in vivo due to lactate usage by nitrate reducing bacteria, including Rothia and Neisseria.
Anie Apriani et al. [48]	Perspective study	Casein CPP-ACP and fluoride varnish effect on saliva pH and caries activity	60 children patients, 30 patients were treated with casein CPP-ACP and 30 patients with fluoride varnish	No statistical differences were found between the two groups.
Rahul G. Padala et al. [86]	Perspective case-control study	to verify the impact of Fluoride-releasing elastic modules on Streptococcus Mutans in oral cavity	Fluoride-releasing elastic modules were placed on brackets 12-11-33 and fluoride-free elastic modules were positioned on brackets 21-22-43 in 30 orthodontic patients. During each appointment, they were analyzed and replaced.	It was reported a considerable reduction was demonstrated (*p* < 0.001) of *S. mutans* count on Fluoride-releasing modules compared to non-fluoridates ones.
M.M. Nascimento et al. [87]	Randomized double-blind clinical trial	to evaluate the trend of plaque metabolic profile using arginine and fluoride.	83 patients’ plaque data were selected on tooth surfaces with and without caries. Taxonomic profiles and analyses on plaque metabolism, arginine catabolism, and acidogenicity were performed.	Biofilm pH homeostasis is improved due to Arginine metabolism. Fluoride improves resistance of enamel and dentine to acidic pH.
Xin Zheng et al. [88]	Clinical Trial	to evaluate Fluoride and Arginine toothpaste impact on oral microbiome.	42 patients divided in 2 groups had to clean their teeth 2 times a day for 3 minutes with and without Fluoride and Arginine toothpaste in 1 month.	Fluoride and arginine work together to maintain an oral microbial balance and prevent dental cavities.
Mine Koruyuc al. [89]	Randomized Clinical Trial	Non-fluoride toothpastes were compared to fluoride toothpastes about clinical, antibacterial, and microbiological effects.	80 patients, aged from 3 to 12 years old, were randomly assigned to four groups and evaluated for 1 month. 2 groups used fluoride toothpaste and 2 groups used non-fluoride toothpaste. These groups were analyzed with statistical analysis	Streptococci Mutans levels decreased statistically significantly (*p* 0.05) in the I, II and III groups during one month; but no considerable variations were found between two groups
Azheen Ali et al. [78]	Randomized double-blind clinical trial	to examine the effect of nano-silver CHX or fluoride mouthwashes on WSLs during orthodontic treatment.	42 patients were separated into three groups (14 patients each) based on the mouthwash (nano-silver, CHX, or fluoride), with 3 months and 6 months of follow-up	WSLs in the nanosilver group is evidently less relevant than in the CHX and fluoride group.

**Table 4 children-09-01014-t004:** Primary studies that examined demineralization [MIH: Molar incisor hypomineralisation].

Authors	Type of Study	Object	Study Design and Timeline	Results
Hernández et al., 2020 [116]	Randomized study	Association of the microbiome and dental molar–incisor hypomineralization	Patients with molar–incisor hypomineralization supragingival samples from healthy and MIH. Marker: 16S rRNA gene	The increased protein content of MIH teeth encourage proteolytic bacteria colonization, promote caries, raise the risk of other oral disorders.
Shishniashvili et al., 2018 [115]	Randomized study	The link between tooth enamel mineralization, oral mucosal diseases, and varying levels of GI imbalance.	Patients with caries, acute or chronic candidiasis and proven dysbiosis.	The degree of dental hard tissue demineralization is influenced by GI microbiota dysbiosis.
Leitão et al., 2018 [117]	Randomized study	The determination of calcium kinetics, binding, and release to/from Streptococcus mutans.	Calcium in *S. mutans* Ingbrit 1600 pellets treated with PIPES buffer, 1 or 10 mM Ca	The relevance of the calcium bacterial reservoir may decrease the power for tooth demineralization when released from the bacterial reservoirs
Senneby et al., 2017 [118]	In vivo study	Intra-individual variability in biofilm acid tolerance between different tooth surfaces and inter-individual variance and acid tolerance stability over time.	Plaque biofilm sampling by 40 adolescents	Biofilm acid tolerance showed short-term stability and low variance between multiple sites in the same individual.
Zhou et al., 2016 [119]	In vivo study	Variations of oral microbial communities by patients with and without caries	Saliva from patients with and without caries	The microbial community structure was influenced by salivary pH and iron content.
Valenti et al., 2021 [120]	Randomized study	Effects of the erbium:yttrio-aluminum-granate (Er:YAG) laser on dental diseases and on bacterial composition	Adults with active deep dental illnesses received CT and Er:YAG therapy.	Er:YAG laser demonstrated to be able to reduce microbial loads, aimed to pediatric and complicated patients

**Table 5 children-09-01014-t005:** Included studies that explored the influence of probiotics and prebiotics on dental remineralization.

Authors	Type of Study	Object	Study Design and Timeline	Results
Nunpan et al. 2019 [70]	Randomized study	The prebiotic’s effect on *Lactobacillus* acidophilus’s inhibition on Streptococcus mutans (A32-2) and prevent dental caries.	*S. mutans* A32-2-*S. mutans* clinical strain obtained from highly active carious individuals	*S. mutans*’ growth rate was significantly inhibited when cocultured with *L. acidophilus* and the appropriate dose of prebiotics.
Campanella et al. 2018 [61]	Randomized double-blinded placebo-controlled pilot study	The therapeutic benefit of oral probiotics on acute oral and respiratory tract infections in pediatric patients	Total 40 subjects with recent oral and respiratory tract infections. The probiotics were compared to placebo.	Major advantages of probiotics in reducing infections in the oral and respiratory tracts without the use of any drugs.

## Data Availability

Not applicable.

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
