# Peer review of "Oralbiotica/Oralbiotics: The Impact of Oral Microbiota on Dental Health and Demineralization: A Systematic Review of the Literature"

_children, 2022, doi:10.3390/children9071014_

Round 1
Reviewer 1 Report
Manuscript of considerable interest for dental professionals, with the focus on changing the oral microbiota based on clinical cases.
Before proceeding with the publication need for a peer review.
Abstract, insert the microbiological variation by citing predisposing bacteria.
Well worded introduction
Materials and methods:
rearrange table 1, very confusing
Increase the resolution of the images of figures 3-4 and table 2-3-and 4
Discussions, also add how the oral microbiota changes in pregnant women and what are the possible correlations with the fetus for the presence of MIH, and add a section of advice on biomimetic hydroxyapatite-based remineralizations.
Conclusions: add proactive remineralization action
Author Response
Report 1
Manuscript of considerable interest for dental professionals, with the focus on changing the oral microbiota based on clinical cases.
Before proceeding with the publication need for a peer review.
Abstract, insert the microbiological variation by citing predisposing bacteria.
ANSWER: We modified the abstract according to the reviewer's suggestion
Well worded introduction
Materials and methods:
rearrange table 1, very confusing
ANSWER: we readjusted the table to match the text for a correct understanding of the search strategy
Increase the resolution of the images of figures 3-4 and table 2-3-and 4.
ANSWER: correction done
Discussions, also add how the oral microbiota changes in pregnant women and what are the possible correlations with the fetus for the presence of MIH, and add a section of advice on biomimetic hydroxyapatite-based remineralizations.
ANSWER: The part on the microbiota in pregnancy and the correlation with MIH in infants and biomimetic hydroxyapatite-based remineralizations has been added.
Conclusions: add proactive remineralization action
ANSWER: the conclusion has been modified according to the reviewer's suggestion

Reviewer 2 Report
The manuscript was written well.
The last paragraph of the introduction should be rewritten
Did the SR register in Prospero?
Tables submitted as figures submit them as tables.
The first column (Ref No) keep next to the authors in the tables.
Google Scholar grey literature
Kindly submit search documents in various databases as supplemental files
The Prisma diagram is not clear; kindly resubmit clear one
PRISMA document for your SR is expected to be submitted by the authors
Limitations of the review should be stated at the end of the manuscript.
Conclusion:
Should be objective-based, rephrase the statement based on the research question
Author Response
Report 2
The manuscript was written well.
The last paragraph of the introduction should be rewritten
ANSWER: the last paragraph has been revised.
Did the SR register in Prospero?
ANSWER: yes it did. full ID for registration: CRD42022331431
Tables submitted as figures submit them as tables.
ANSWER: fixed
The first column (Ref No) keep next to the authors in the tables.
ANSWER: Checked and corrected.
Google Scholar grey literature
ANSWER: google scholar has been used only as secondary source
Kindly submit search documents in various databases as supplemental files
ANSWER: we will provide the documents in the supplemental files. We have attached the excel file of the identification phase and the excel file of the eligibility phase. In addition, we have attached a folder with PDF files of the articles selected for eligibility.
The Prisma diagram is not clear; kindly resubmit clear one
ANSWER: the diagram has been improved
PRISMA document for your SR is expected to be submitted by the authors
ANSWER: the authors are going to attach in the supplemental files the PRISMA prospero document
Limitations of the review should be stated at the end of the manuscript.
Conclusion:
Should be objective-based, rephrase the statement based on the research question
ANSWER: the conclusion section has been revised accordingly by adding the limits and making our research topic more punctual to the reader. The authors thank the reviewer for the suggestions.

Reviewer 3 Report
The authors conducted a literature review on the effects of oral microflora on dental health and demineralization.The background, discussion, and methodology are well written.However, one point about methodology needs to be confirmed and explained, which is described below.
The reporting is done according to PRISMA's guidelines and quality is assured. However, several In vivo studies are included in the included papers.
In general, the results of in vivo and clinical studies cannot be homogeneously compared and examined, but I feel a strong sense of discomfort that this is being discussed as the same level of research.
This is a topic that needs to be properly explained and in some cases the inclusion and exclusion criteria of the paper needs to be confirmed. Please consider this matter.
Thank you very much.
Author Response
Report 3
The authors conducted a literature review on the effects of oral microflora on dental health and demineralization.The background, discussion, and methodology are well written.However, one point about methodology needs to be confirmed and explained, which is described below.
The reporting is done according to PRISMA's guidelines and quality is assured. However, several In vivo studies are included in the included papers.
In general, the results of in vivo and clinical studies cannot be homogeneously compared and examined, but I feel a strong sense of discomfort that this is being discussed as the same level of research.
This is a topic that needs to be properly explained and in some cases the inclusion and exclusion criteria of the paper needs to be confirmed. Please consider this matter.
Thank you very much.
ANSWER. We thank the reviewer for his concern and suggestion. The authors conducted their research strategy by considering all in vivo and clinical studies because, as it has been found in the literature, the in vivo method is almost equivalent to clinical trials[1](page 1). This led to a more comprehensive and complete list to describe the state of the art of the topic of this systematic review.
If the reviewer has further suggestions to make the research strategy more punctual and precise , the authors are going to comply accordingly.
- Sekar, P.; S, N.; Desai, V. Recent Progress in in Vivo Studies and Clinical Applications of Magnesium Based Biodegradable Implants – A Review. J. Magnes. Alloys 2021, 9, 1147–1163, doi:10.1016/j.jma.2020.11.001.

Round 2
Reviewer 2 Report
All the queries have been addressed
Now the manuscript looks better in shape